# Developing a Simple Score for Diagnosis of Acute Cholecystitis at the Emergency Department

**DOI:** 10.3390/diagnostics12092246

**Published:** 2022-09-17

**Authors:** Saowaluck Faikhongngoen, Boriboon Chenthanakij, Borwon Wittayachamnankul, Phichayut Phinyo, Wachira Wongtanasarasin

**Affiliations:** 1Department of Emergency Medicine, Faculty of Medicine, Chiang Mai University, Chiang Mai 50200, Thailand; 2Department of Family Medicine, Faculty of Medicine, Chiang Mai University, Chiang Mai 50200, Thailand; 3Center for Clinical Epidemiology and Clinical Statistics, Faculty of Medicine, Chiang Mai University, Chiang Mai 50200, Thailand; 4Department of Emergency Medicine, UC Davis School of Medicine, Sacramento, CA 95817, USA

**Keywords:** acute cholecystitis, prediction, acute abdominal pain, emergency department

## Abstract

We aim to develop a diagnostic score for acute cholecystitis that integrates symptoms, physical examinations, and laboratory data to help clinicians for timely detection and early treatment of this disease. We retrospectively collected data from our database from 2010 to 2020. Patients with acute abdominal pain who underwent an ultrasound or computed tomography (CT) scan at the emergency department (ED) were included. Cases were identified by pathological, CT, or ultrasound reports. Non-cases were those who did not fulfill any of these criteria. Multivariable regression analysis was conducted to identify predictors of acute cholecystitis. The model included 244 patients suspected of acute cholecystitis. Eighty-six patients (35.2%) were acute cholecystitis confirmed cases. Five final predictors remained within the reduced logistic model: age < 60, nausea and/or vomiting, right upper quadrant pain, positive Murphy’s sign, and AST ≥ two times upper limit of normal. A practical score diagnostic performance was AuROC 0.74 (95% CI, 0.67–0.81). Patients were categorized with a high probability of acute cholecystitis at score points of 9–12 with a positive likelihood ratio of 3.79 (95% CI, 1.68–8.94). ED Chole Score from these five predictors may aid in diagnosing acute cholecystitis at ED. Patients with an ED Chole Score >8 should be further investigated.

## 1. Introduction

Acute abdominal pain (AAP) is a common symptom for patients who visit the Emergency Department (ED) worldwide [1]. Acute cholecystitis is one of the common causes of AAP. The incidence of acute cholecystitis was estimated at around 3–11% of the general population [1]. Additionally, it is associated with a mortality rate of 0.8% [1]. However, acute right upper quadrant (RUQ) pain has a wide range of differential diagnoses, including acute cholecystitis, acute cholangitis, liver abscess, and acute hepatitis [2]. Hence, an accurate and timely diagnosis of patients with acute RUQ pain is crucial [3,4].

The Tokyo Guidelines (TG) were initially published in 2007 and modified in 2018. They include clinical and radiologic diagnostic criteria for acute cholecystitis developed to handle the issue regarding the optimal criteria for clinical diagnosis [5,6]. According to the latest version of TG, the diagnosis of acute cholecystitis must include all of the following information: history and physical examinations, laboratory results, and radiographic findings, such as ultrasound, computed tomography (CT), and magnetic resonance imaging (MRI) [6]. A validation study of the TG reported a sensitivity of 91.2% and specificity of 96.9% [7]. However, approximately 28% of these patients were diagnosed with acute cholecystitis without fever and leukocytosis, especially in the elderly [8]. Ultrasound is recommended for all suspected acute cholecystitis patients, since it is a simple and non-invasive procedure [3]. Nevertheless, there are limitations because this score depends on the clinician’s experience and expertise [9,10]. The accuracy of the ultrasound alone has been debated, with a sensitivity ranging from 48% to 94% [11]. As a result, the TG requires a radiographic examination to diagnose acute cholecystitis, especially an ultrasound. However, radiologists or emergency physicians may not be available in all circumstances, especially in the community or local hospitals. This study aims to develop a practical diagnostic tool for acute cholecystitis that integrates clinical signs and symptoms, physical examinations, and primary laboratory data to aid physicians in community hospitals for timely detection and early treatment.

## 2. Materials and Methods

This study was prepared and reported according to the transparent reporting of a multivariable prediction model for individual prognosis or diagnosis [12].

### 2.1. Study Design and Selection of Participants

We performed diagnostic prediction research and practical diagnostic score development [13]. We retrospectively collected the data from the *Chiang Mai University Hospital* Electronic Medical Record database system. Chiang Mai University Hospital is a university hospital that consists of 1500 patient beds, 151 intensive care units (ICU) and sub-ICUs, 28 operating rooms, and consultant doctors, all specialists. Of all, the general surgical ward where acute cholecystitis patients are admitted has up to 200 beds. Adults (age ≥ 18 years) who intended to be diagnosed with acute cholecystitis by emergency physicians from their initial clinical presentations and visiting our ED between January 2010 and December 2020 were collected and enrolled in the study. Patients with traumatic mechanisms and pregnant women were excluded.

### 2.2. Data Collection

Patients who intended to be diagnosed with acute cholecystitis (AAP and were performed either US by attending radiologists or CT scan at the ED) were enrolled in this study. After a structured review of hospital medical records, trained abstractors recorded the data. Abstractors were doctors and paramedics with at least two years of practice at the ED. If there were more than one piece of information in each variable, the only first recorded information was used to represent that variable. For example, vital signs were the first obtained from the patient at the ED. Each eligible record was reviewed and assessed by one abstractor. Two authors (S.F. and W.W.) randomly checked some papers to find any errors in the abstraction process. Any disagreement arising from the reviewers was resolved by consensus. Missing, confusing, and ambiguous chart elements were coded as unknown variables.

The relevant data were selected and entered into REDcap (Vanderbilt University, Nashville, TN, USA). Specifically, we collected the data on four components: (1) demographic data (gender and age), (2) clinical characteristics (fever, nausea/vomiting, jaundice, and RUQ pain), (3) physical examinations (body temperature, pulse rate, blood pressure, respiratory rate, rebound tenderness at RUQ, and Murphy’s sign), and (4) laboratory findings (white blood cell count, percentage of neutrophil, absolute neutrophil count, platelet count, alanine aminotransferase, aspartate aminotransferase, alkaline phosphatase, total bilirubin, and direct bilirubin). All subjective variables were based on the history obtained from the patient. For example, fever was considered positive if patients reported it. Jaundice is defined as a condition in which a yellowish tinge appears on the sclerae, mucous membrane, or skin. We chose rebound tenderness at RUQ because it represented a sign of RUQ-localized peritonitis, which is associated with inflammation or infection of a gallbladder. If rebound tenderness and Murphy’s sign were not documented as either positive or negative, they were coded as unknown variables.

### 2.3. Confirmation of Cases

Clinically suspected patients were defined as “Acute cholecystitis confirmed cases” if one of the following criteria were met: (1) pathological report, (2) CT report, and (3) US report by radiologist. Patients who did not fulfill any criteria were classified as “non-cases.” In addition, the confirmation of diagnosis other than acute cholecystitis in non-cases patients was not done.

### 2.4. Statistical Analysis and Sample Size Estimation

Since there is currently no standard approach for estimating study size to develop clinical prediction rules, we reviewed the unpublished data and patient records comparing the clinical characteristics of acute cholecystitis confirmed cases and non-cases at our hospital in 2018. The proportion of patients with RUQ was 0.37 and 0.63 for confirmed cases and non-cases of acute cholecystitis. Using the comparison of the two proportions approach, 21 confirmed cases and 34 non-cases were needed to achieve a statistical power of 80 percent and a two-sided alpha error of 0.05. Furthermore, much literature suggested a 10-events-per-variable rule, including the TRIPOD statements for reporting clinical prediction rules development [14]. For our study, as we planned to include at least five potential predictors within the finished model, at least 50 confirmed cases were required for model derivation. With a confirmed cases:non-cases ratio of 1:4 [15], this study planned to recruit at least 250 patients (50:200).

### 2.5. Model Development and Internal Validation

The model was built using complete-case analysis with no data imputation. The multivariable logistic regression model included all clinically relevant parameters to explore significant predictors of acute cholecystitis. Backward elimination was performed based on statistical significance from *p*-values for each predictor and the total predictive performance of the model as measured by the area under the receiver operating characteristic curve (AuROC). Non-contributing factors with high *p*-values and a minor magnitude of effect (odds ratios close to 1.00) were initially removed from the regression model. After removing each predictor, we used the AuROC to assess model diagnostic performance. Removing the predictor decreased AuROC; it was re-added to the model. The steps were repeated until all of the remaining predictors in the model had a *p*-value less than 0.10, and the AuROC of the reduced model was well preserved. Some important clinical parameters that might not have statistical significance were added to the full model based on the authors’ discussion. AuROC curves were used to measure discrimination and calibrate the final reduced model. According to Hosmer and Lemeshow, acceptable, good, and outstanding AuROC were defined as an AuROC of 0.70–0.80, 0.80–0.90, and above 0.90, respectively [16]. To demonstrate the visual inspection of the agreement between the derived score (ED Chole Score) and the actual (observed) risk, we generated the calibration plot by comparing the overall score sum to the observed proportion of confirmed cases within each score strata. Additionally, the Hosmer–Lemeshow goodness-of-fit test was calculated.

### 2.6. Score Derivation and Validation

Each predictor in the final model received a score based on its multivariable logistic odds ratio (mOR). The lowest mOR of all predictors was used as a denominator in score transformation, while others were used as numerators. Following the mOR division, the products were rounded up to non-decimal numbers. Each patient in the development cohort received a score. For the score-based logistic regression model, discrimination and calibration were measured and calibrated similarly. The score was classified into three risk groups (low, moderate, and high probability). Each risk category’s sensitivity, specificity, and positive likelihood ratio (PLR) will also be displayed.

## 3. Results

### 3.1. Participants

Of all 257 patients intended to be diagnosed with acute cholecystitis, 244 patients were included in the model development. Moreover, 86 (35.2%) were confirmed acute cholecystitis cases (Figure 1).

Pathological reports confirmed that radiological reports confirmed 52 (60.5%) and 34 (39.5%) cases. The mean age of the whole cohort was 51.5 ± 18.3 years. The acute cholecystitis group had younger age than the non-cases group (47.9 vs. 53.4 years, OR 0.98 (95% CI, 0.97–1.00), *p* = 0.03). For clinical characteristics, confirmed cases had much RUQ pain compared to non-cases (81.4% vs. 67.7%, OR 2.09 (95% CI, 1.10–3.94), *p* = 0.02). Most confirmed cases presented with nausea and/or vomiting (51.2% vs. 30.4%, OR 2.40 (95% CI, 1.40–4.13), *p* = 0.002). Most of the physical examinations between both groups were similar. The laboratory findings which showed statistical significance were serum aspartate aminotransferase level (median 58 U/L (IQR 21-136) vs. 28 U/L (IQR 19-64), *p* = 0.003, for cases and non-cases, respectively) and serum alanine aminotransferase level (median 38 U/L (IQR 21-141) vs. 27 U/L (IQR 17-53), *p* = 0.004, for cases and non-cases, respectively). The remaining diagnostic variables without statistically significant differences from the univariable comparison are shown in Table 1.

### 3.2. Model Development and Validation

All variables were included in multivariable logistic regression for model development and score generation (Table 2). Sequential elimination was performed based on statistical significance and model diagnostic performance. After careful consideration based on both clinical and statistical significance, five final predictors remained within the reduced logistic model, including age less than 60 years old, nausea and/or vomiting, RUQ pain, positive Murphy’s sign, and serum aspartate aminotransferase (AST) ≥two times upper limit of normal (ULN). Although RUQ pain and positive Murphy’s sign had a *p*-value of more than 0.1, we decided to include these two factors in the final model, since they represented signs of RUQ peritonitis and should be concerned for acute cholecystitis in clinical practice. The score was derived by dividing the larger mOR by the smallest mOR, resulting in 12 points. The newly developed diagnostic scoring scheme was named “ED Chole Score.” Age < 60 received 2 points, nausea and/or vomiting received 3 points, RUQ pain received 2 points, positive Murphy’s sign received 1 point, and AST 2xULN received 4 points (Table 3). Based on the PLR and *p*-value, three pre-specified cut-off points for score dichotomization were compared (Table 4). The PLR of scores 0 to 4 for diagnosis of acute cholecystitis was 0.32 (95% CI, 0.15–0.64, *p* < 0.001). The PLR of scores 9 to 12 for diagnosis of acute cholecystitis was 3.79 (95% CI, 1.69–8.94, *p* < 0.001). The ED Chole Score diagnostic performance was considered satisfactory at the AuROC of 0.74 (95% CI, 0.67–0.81, Figure 2). The Hosmer–Lemeshow goodness of fit test for model calibration was insignificant for the diagnosis (*p* = 0.8612). Furthermore, the calibration plot, which visualized the agreement between the predicted and the actual (observed) risk of acute cholecystitis confirmed cases, demonstrated satisfactory calibration (Figure 3).

## 4. Discussion

Clinical signs and symptoms were indistinguishable in the early stages of acute cholecystitis and did not provide sufficiently significant diagnostic value. This study identified five final predictor variables: clinical, physical examinations, and basic routinely available laboratory investigations. Age less than 60 years old, nausea and/or vomiting, RUQ pain, positive Murphy’s sign, and AST ≥2xULN were included in the multivariable model for diagnosing acute cholecystitis patients.

The TG has been widely endorsed since 2007 [5,17]. In 2013, the criteria were revised to improve sensitivity and specificity [6]. Five years later, the latest version of the criteria was launched as TG 2018 [7]. Those criteria consisted of three parts: local signs of inflammation, systemic signs of inflammation, and imaging findings [7]. Three predictor variables were needed from the patient’s clinical presentation, including RUQ pain, positive Murphy’s sign, and fever. Two types of laboratory investigation (elevated WBC count and elevated CRP) were represented for systemic signs of inflammation. For imaging, findings of acute cholecystitis (such as fat stranding, pericholecystic fluid, gallbladder wall thickening, and sonographic positive Murphy’s sign) were also necessary to fulfill the criteria [7]. All the criteria were based on acutely ill patients with the typical clinical syndrome. They also required advanced laboratory testing, such as CRP and imaging findings, which were usually unavailable, especially in health resource-limited areas.

Advanced age was considered one of the risk factors for acute cholecystitis, and most cases (50–70%) were aged > 65 years old [7,18]. However, our study revealed a difference. Our finding may result from the fact that we included only patients with documented AAP. Elderly patients might incorrectly identify the pain sensation and location of the pain [19]. Additionally, some elderly patients mistakenly believe pain is a normal aging process [19]. In this study, the accuracy of Murphy’s sign has been debated, with some authors reporting high sensitivity and predictive values [20]. There are no sources in the current document. However, the other authors reported Murphy’s sign’s high specificity but low sensitivity [6]. The conclusion is that Murphy’s sign is strongly suggestive (but not diagnostic) of acute cholecystitis [21,22,23]. However, its absence does not exclude the disease. For example, Murphy’s sign is frequently absent in the case of gangrenous cholecystitis, possibly due to the denervation of the gallbladder wall secondary to ischemic changes [2]. Five potential predictors constituted a total score of twelve with acceptable diagnostic performance according to the AuROC. In a practical score implementation, we chose cut-offs at 0–4 and 9–12 points because the PLR was within an acceptable margin and not too low to cause a diagnostic error. The score was designed to be used in undifferentiated RUQ pain patients intended to be diagnosed with acute cholecystitis, so the patients’ signs and symptoms must be relevant. In addition, laboratory tests were also used in diagnosing acute cholecystitis. They can represent the evidence of systemic inflammations and aid clinicians in giving a diagnosis promptly. Our score used a serum AST because of its significant diagnostic performance and accessibility. Patients who were categorized as low risk should be closely monitored for disease progression. In contrast, patients who were categorized as high risk could be offered a choice of antibiotic and prompt surgeon consultation for proper management. Since radiologists were generally not persisted in the community or local hospitals, clinicians could exploit the score as a guide in initiating early surgeon consultation by leaving out imaging findings and still preserving a diagnostic performance of 0.74.

This study was primarily designed for patients who presented with undifferentiated AAP and were suspected of having acute cholecystitis. The domain of patients in our study was shifted from clear diagnosis from pathology report as in other derived criteria to patients who intended to be diagnosed with acute cholecystitis with undifferentiated symptoms. We believe that early diagnosis and timely treatment in a suspected group of patients could delay disease progression and reduce mortality.

There are several limitations in this study. The development cohort consisted of patients all treated at a single university hospital, and the results may not be generalized to other settings. Second, as this study extends from the TG, some parameters (history and physical examinations) were derived from the TG data. Consequently, the data that might have impacted this study, such as an underlying disease, were not collected. For example, patients with diabetes tended to have more severe clinical features than non-diabetic patients [24]. Besides, the previous article focusing on clinical features of acute cholecystitis found that >2 mg/mL of serum creatinine was observed in diabetic patients, which is more than in non-diabetes [24]. Furthermore, because several different levels of doctors are working in the ED, some physical examination requires both skill and experience in interpreting. This is a limitation of the retrospective study design and the local practice. Third, several variables were not recorded and included in this study (i.e., duration of the pain, possible triggers for the pain, body mass index). In addition, medical records are sometimes completed after a diagnosis is made, which means that the diagnosis could have influenced the documented findings. Besides, this study included patients who had AAP and underwent US or CT at ED; therefore, some patients with other symptoms were excluded, which might result in selection bias. Furthermore, although acute cholecystitis after cholecystectomy is rare, some articles highlighted this condition and should be cautiously considered [25,26]. Recurrence of symptoms comparable to cholecystitis following cholecystectomy should not be referred to entities such as the post-cholecystectomy syndrome until the condition has been thoroughly evaluated [25]. Moreover, some patients with negative ultrasound findings did not undergo CT or further investigations, resulting in a false negative sample. Notably, the data collection duration overlapped with the onset of the COVID-19 pandemic. COVID-19 can cause several non-specific symptoms, such as fever and abdominal pain, and could contribute to selection bias [27]. That illness has famously caused mild to moderate organ dysfunction and might result in abnormal laboratory values (i.e., WBC, AST) [28]. However, we reviewed all patients who presented during 2020 and had not found any positive COVID-19 cases in our cohort. Finally, this study was conducted in a clinical prediction development manner. We used our cohort as a developing population and analyzed the internal validity of this score only for our cohort. As a result, an external validation study is needed to evaluate generalizability before the score is endorsed for real-world clinical practice.

## 5. Conclusions

To conclude, we developed a practical prediction score to aid clinicians in diagnosing acute cholecystitis in patients presenting to the ED with acute undifferentiated abdominal pain without radiographic examination. With an acceptable range of diagnostic performance, we recommend using this simple score for timely diagnosis and early management for patients suspected of acute cholecystitis, especially in a setting with limited further investigations. Patients with an ED Chole Score of 0–4 could be managed considerately. In contrast, patients with an ED Chole Score of more than eight should be further investigated for acute cholecystitis. In addition, it is advised that a prospective external validation study with a larger sample size should be conducted before applying this score in clinical practice.

## Figures and Tables

**Figure 1 diagnostics-12-02246-f001:**
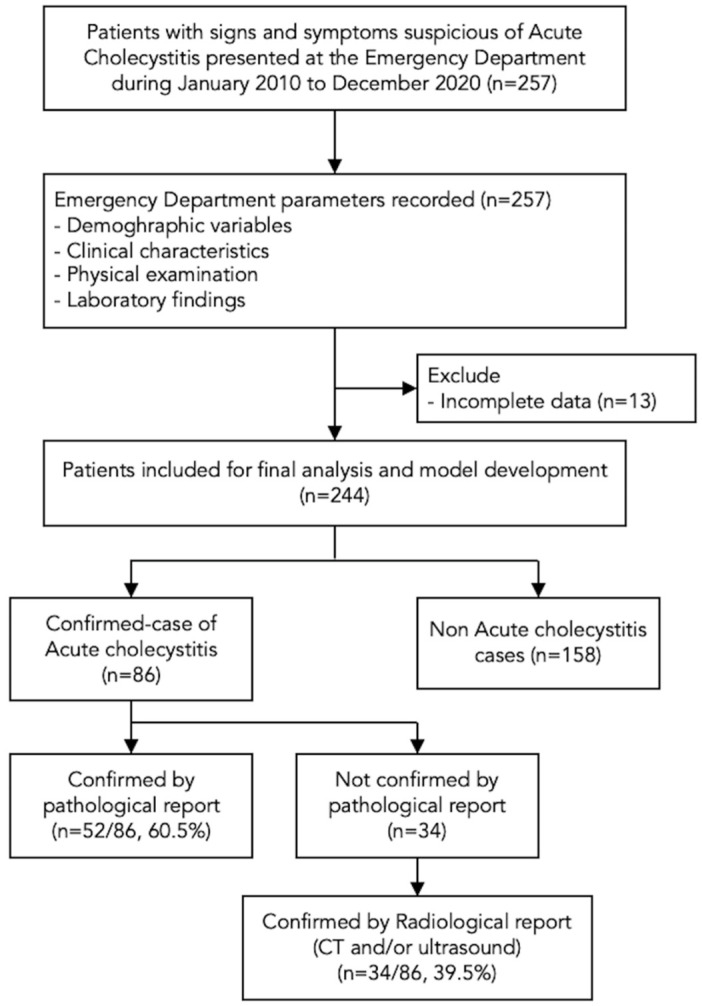
Study flowchart.

**Figure 2 diagnostics-12-02246-f002:**
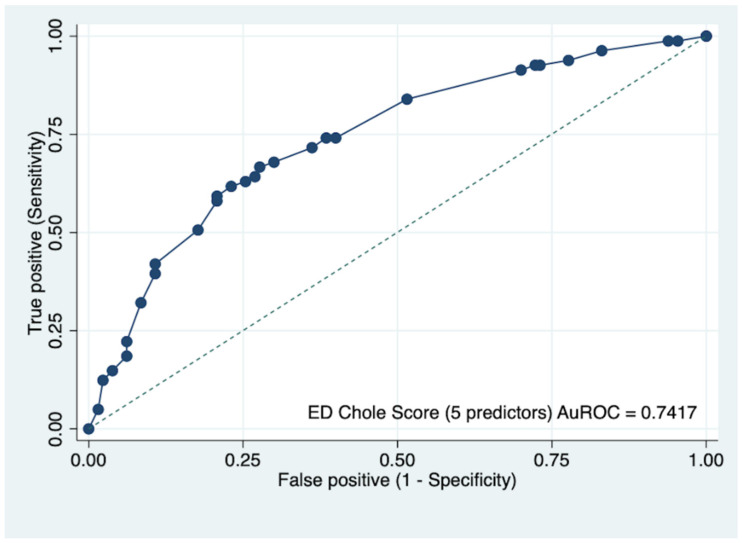
Receiver operating characteristic curve in the diagnostic prediction of acute cholecystitis of ED Chole Score within the development cohort.

**Figure 3 diagnostics-12-02246-f003:**
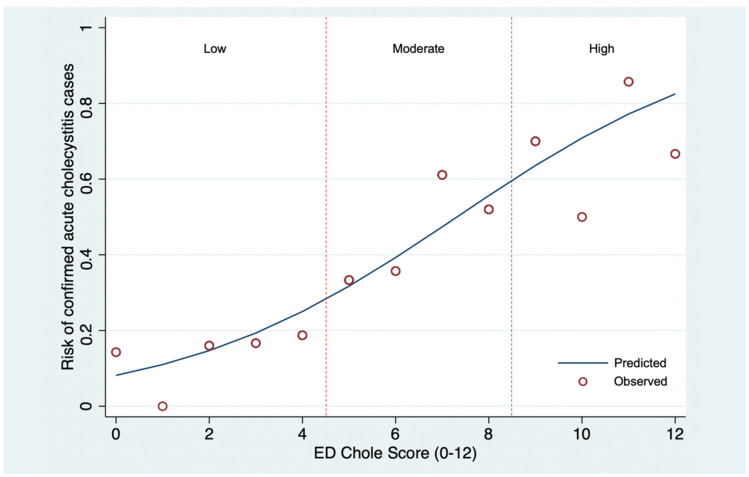
Calibration plots visualizing agreement between the derived score (ED Chole Score) and the actual (observed) risk. The blue line depicts the predict risk of acute cholecystitis confirmed cases. The red circle defines the observed risk of acute cholecystitis confirmed cases.

**Table 1 diagnostics-12-02246-t001:** Baseline demographic variables, clinical characteristics, physical examinations, and initial laboratory investigations of the derivation cohort; comparison of confirmed cases and non-cases of acute cholecystitis.

Variables	Missing Data, n (%)	Confirmed Cases, n (%) (n = 86)	Non-Cases, n (%) (n = 158)	Crude OR (95% CI)	*p*-Value	AuROC (95% CI)
**Demographic data**						
Male	0 (0)	48 (55.8)	88 (55.7)	1.00 (0.59–1.71)	0.99	50.06 (43.51–56.61)
Age, years (mean ± SD)	0 (0)	47.9 ± 15.9	53.4 ± 19.2	0.98 (0.97–1.00)	0.03	42.10 (34.86–49.35)
Age < 60 years old		67 (77.9)	101 (63.9)	1.99 (1.09–3.64)	0.03	56.99 (51.20–62.78)
**Clinical characteristics**						
Fever	0 (0)	14 (16.3)	41 (26.0)	0.55 (0.28–1.09)	0.09	45.16 (39.95–50.38)
Nausea/vomiting	0 (0)	44 (51.2)	48 (30.4)	2.40 (1.40–4.13)	0.002	60.39 (53.98–66.81)
Jaundice	0 (0)	13 (51.1)	20 (12.7)	1.23 (0.58–2.61)	0.59	51.23 (46.62–55.84)
RUQ pain	0 (0)	70 (81.4)	107 (67.7)	2.09 (1.10–3.94)	0.02	56.84 (51.32–62.36)
**Physical examinations**						
Body temperature, °C (mean ± SD)	29 (11.9)	36.9 ± 0.9	37.1 ± 1.0	0.80 (0.60–1.08)	0.14	43.96 (37.98–49.94)
<36 °C		9 (37.5)	15 (62.5)	0.95 (0.39–2.31)	0.91	
36–38 °C		60 (38.7)	95 (61.3)	Ref	Ref	
>38 °C		7 (19.4)	29 (80.6)	0.38 (0.16–0.93)	0.03	
Pulse rate, per mins (mean ± SD)	7 (2.9)	85.9 ± 21.0	89.3 ± 18.7	0.99 (0.98–1.00)	0.20	48.71 (42.78–54.64)
>60 bpm		6 (54.6)	5 (45.5)	2.18 (0.64–7.45)	0.21	
60–100 bpm		61 (35.5)	111 (64.5)	Ref	Ref	
>100 bpm		16 (29.6)	38 (70.4)	0.77 (0.40–1.49)	0.43	
SBP, mmHg (mean ± SD)	8 (3.3)	128.7 ± 25.4	131.6 ± 23.8	1.00 (0.98–1.01)	0.38	52.36 (49.26–55.46)
≥90 mmHg		76 (33.6)	150 (66.4)	Ref	Ref	
<90 mmHg		6 (60.0)	4 (40.0)	2.96 (0.81–10.81)	0.10	
DBP, mmHg (mean ± SD)	8 (3.3)	77.1 ± 18.5	78.5 ± 16.7	1.00 (0.98–1.01)	0.54	53.34 (48.59–58.10)
≥60 mmHg		68 (82.9)	138 (89.6)	Ref	Ref	
<60 mmHg		14 (17.1)	16 (10.4)	1.78 (0.82–3.85)	0.15	
Respiratory rate, per mins (mean ± SD)	23 (9.4)	20.7 ± 4.4	20.3 ± 4.6	1.02 (0.96–1.08)	0.53	49.17 (46.30–52.04)
<30 per mins		74 (96.1)	136 (94.4)	Ref	Ref	
≥30 per mins		3 (3.9)	8 (5.6)	0.69 (0.18–2.68)	0.59	
Rebound tenderness at RUQ	0 (0)	7 (8.1)	7 (4.4)	1.91 (0.65–5.64)	0.24	51.85 (48.53–55.18)
Positive Murphy’s sign	0 (0)	38 (44.2)	57 (36.1)	1.40 (0.82–2.40)	0.22	54.06 (47.58–60.53)
**Laboratory findings**						
WBC/μL (median, IQR)	13 (5.3)	11,021 (8,000–15,560)	11,170 (8,180–14,025)	1.00 (1.00–1.00)	0.28	51.61 (44.90–58.32)
<12,000/μ		45 (54.2)	85 (57.4)	Ref	Ref	
≥12,000/μ		38 (45.8)	63 (42.3)	1.14 (0.66–1.96)	0.64	
Neutrophil, % (mean ± SD)	13 (5.3)	75.2 ± 13.2	72.4 ± 14.7	1.01 (0.00–1.03)	0.16	51.21 (44.67–57.75)
Neutrophil count ≥ 80%		38 (44.2)	66 (41.8)	1.10 (0.65–1.88)	0.72	
Absolute neutrophil count/μL (median, IQR)	13 (5.3)	8,257.4 (1,055.7–22,070.1)	7,831.7 (4,477.9–10,795.7)	1.00 (1.00–1.00)	0.11	55.20 (47.62–62.78)
Platelet/μL (median, IQR)	15 (6.1)	242,000 (192,000–297,000)	245,000 (192,000–313,000)	1.00 (1.00–1.00)	0.37	47.62 (39.93–55.33)
AST (median, IQR)	33 (13.5)	58 (21–136)	28 (19–64)	1.00 (1.00–1.01)	0.003	64.84 (58.43–71.26)
<2xULN		42 (51.9)	106 (81.5)	Ref	Ref	
≥2xULN		39 (48.2)	24 (18.5)	4.10 (2.20–7.64)	<0.001	
ALT (median, IQR)	32 (13.1)	38 (21–141)	27 (17–53)	1.00 (1.00–1.01)	0.004	57.51 (51.36–63.65)
<2xULN		54 (66.7)	107 (81.7)	Ref	Ref	
≥2xULN		27 (33.3)	24 (18.3)	2.23 (1.18–4.23)	0.01	
Alkaline phosphatase (median, IQR)	36 (14.8)	98 (70–169)	92 (67–148)	1.00 (1.00–1.00)	0.39	49.93 (44.38–55.49)
<2xULN		61 (81.3)	108 (81.2)	Ref	Ref	
≥2xULN		14 (18.7)	25 (18.8)	0.99 (0.48–2.05)	0.98	
Total bilirubin (median, IQR)	31 (12.7)	1.02 (0.55–2.48)	0.68 (0.40–1.42)	1.02 (0.95–1.10)	0.54	57.01 (50.74–63.28)
<2 mg/dL		52 (65.8)	107 (79.9)	Ref	Ref	
≥2 mg/dL		27 (34.2)	27 (20.2)	2.06 (1.10–3.86)	0.02	
Direct bilirubin (median, IQR)	30 (12.3)	0.47 (0.20-.1.22)	0.24 (0.16–0.64)	1.03 (0.91–1.17)	0.63	53.16 (47.73–58.60)
<1.5 mg/dL		63 (78.8)	114 (85.1)	Ref	Ref	
≥1.5 mg/dL		17 (21.2)	20 (14.9)	1.54 (0.75–3.15)	0.24	

Abbreviations: ALT, alanine aminotransferase; AST, Aspartate aminotransferase; AuROC, the area under the receiver operating characteristic; CI, confidence interval; DBP, diastolic blood pressure; IQR, interquartile range; OR, odds ratio; RUQ, right upper quadrant; SBP, systolic blood pressure; SD, standard deviation; ULN, upper limit of normal.

**Table 2 diagnostics-12-02246-t002:** Multivariable logistic regression analysis.

Variables	Full Model mOR	95% CI	*p*-Value	Reduced Model mOR	95% CI	*p*-Value
**Demographic data**						
Male	0.87	0.39–1.95	0.74			
Age, years	0.97	0.94–1.01	0.14			
Age < 60 years old	0.78	0.18–3.40	0.74	2.02	1.00–4.06	0.05
**Clinical characteristics**						
Fever	0.77	0.24–2.50	0.66			
Nausea/vomiting	1.49	0.66–3.37	0.34	2.66	1.42–4.99	0.002
Jaundice	0.76	0.16–3.66	0.73			
RUQ pain	1.17	0.42–3.21	0.77	1.85	0.84–4.05	0.13
**Physical examinations**						
Body temperature, °C	0.81	0.48–1.36	0.43			
Pulse rate, per mins	0.98	0.96–1.01	0.22			
SBP, mmHg	0.99	0.97–1.01	0.38			
DBP, mmHg	1.00	0.98–1.03	0.83			
Respiratory rate, per mins	1.04	0.93–1.15	0.50			
Rebound tenderness at RUQ	0.76	0.15–3.88	0.75			
Positive Murphy’s sign	1.49	0.62–3.58	0.37	1.12	0.58–2.14	0.74
**Laboratory findings**						
WBC/μL	1.00	1.00–1.00	0.92			
Neutrophil, %	1.04	0.97–1.11	0.26			
Absolute neutrophil count/μL	1.00	1.00–1.00	0.73			
Platelet/μL	1.00	1.00–1.00	0.34			
AST	1.00	1.00–1.00	0.83			
AST ≥ 2xULN	8.28	3.20–31.14	0.002	4.21	2.19–8.08	<0.001
ALT ≥ 2xULN	0.84	0.23–3.10	0.79			
Alkaline phosphatase	1.00	1.00–1.00	0.10			
Total bilirubin	0.95	0.67–1.34	0.77			
Direct bilirubin	1.03	0.58–1.83	0.92			
Constant (intercept)	3715.24			0.09		

Abbreviations: ALT, alanine aminotransferase; AST, Aspartate aminotransferase; CI, confidence interval; DBP, diastolic blood pressure; mOR, multivariable odds ratio; RUQ, right upper quadrant; SBP, systolic blood pressure.

**Table 3 diagnostics-12-02246-t003:** A multivariable logistic model with score transformation via multivariable odds ratios.

Predictors	mOR	95% CI	*p*-Value	Score
Age < 60 years old	2.02	1.00–4.06	0.05	2
Nausea/vomiting	2.66	1.42–4.99	0.002	3
RUQ pain	1.85	0.84–4.05	0.13	2
Positive Murphy’s sign	1.12	0.58–2.14	0.74	1
AST ≥ 2xULN	4.21	2.19–8.08	<0.001	4
Constant	0.09			

Abbreviations: AST, Aspartate aminotransferase; CI, confidence interval; mOR, multivariable odds ratio; RUQ, right upper quadrant; ULN, upper limit of normal.

**Table 4 diagnostics-12-02246-t004:** Score categorization and likelihood ratio of positive (LHR+) in ED Chole Score.

Probable Categories	Score	Confirmed Cases, n (%) (n = 81)	Non-Cases, n (%) (n = 130)	LHR+	(95% CI)	*p*-Value
ED Chole Score						
Low	0–4	13 (16.0)	65 (50.0)	0.32	0.15–0.64	<0.001
Moderate	5–8	42 (51.9)	54 (41.5)	1.25	0.74–2.09	0.37
High	9–12	26 (32.1)	11 (8.5)	3.79	1.69–8.94	<0.001
Mean ± SD		7.1 ± 2.7	4.7 ± 2.6			<0.001

Abbreviations: CI, confidence interval; SD, standard deviation.

## Data Availability

Data cannot be shared publicly because it was not formally approved by the Ethics Committee. Anonymous data are available from the Faculty of Medicine, Chiang Mai University Research Administration Office for researchers who meet the criteria for access to confidential data.

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
