# Peer review of "Developing a Simple Score for Diagnosis of Acute Cholecystitis at the Emergency Department"

_diagnostics, 2022, doi:10.3390/diagnostics12092246_

Round 1

Reviewer 1 Report

The study is interesting and well conducted. I have these two major concerns:

1) It is completely unclear to me how the validation analysis was performed. Could you explain better? also, data from validation analysis seem missing

2) The model should be also calibrated...a calibration curve should be added and commented

3) In the methods, some statistical reference from previous studies with similar methodology should be added. For example PMID: 27005802

Author Response

Response to Reviewer 1

We want to thank the editor and reviewers for their helpful suggestions. We have revised the manuscript and hope that this revised version has met the publication criteria. Changes were marked with a track change function in MS Word.

The study is interesting and well conducted. I have these two major concerns:

1) It is completely unclear to me how the validation analysis was performed. Could you explain better? also, data from validation analysis seem missing

Answer: Thank you for asking this point. This study was conducted in a clinical prediction development manner. We used our cohort as a developing population and analyzed the internal validity of this score only for our cohort. We admitted that the clinical prediction score should be validated before being utilized in clinical practice; however, it is out of our scope in this present study. As a result, an external validation study is needed to evaluate the generalizability before the score is endorsed for real-world clinical practice. We indicated this point as the study limitation and revised the ‘Methods’ section, the limitation paragraph in the ‘Discussion’ section,’ and ‘Conclusion’ section (Page 3 Line 118-137, Page 11 Line 324-327 and 337-338).

2) The model should be also calibrated...a calibration curve should be added and commented

Answer: We appreciate the reviewer’s comment on this point. Clinical score performance should be evaluated regarding discrimination, calibration, and clinical utility. The AuROC curve was presented to assess the discriminative capacity of the obtained score. In addition, we created the model calibration plot to demonstrate the visual inspection of the agreement between the derived scores (ED Chole Score) and the actual (observed) risk. Also, we used the Hosmer-Lemeshow goodness-of-fit test to assess the calibration performance. In this study, Hosmer–Lemeshow goodness-of-fit test was insignificant (p = 0.8612). We have revised this point in the ‘Methods’ and ‘Results’ section in the revised manuscript and added Figure 3 to illustrate the score performance in terms of calibration aspect (Page 3 Line 132-137, Page 7 Line 195-199, and Figure 3)

3) In the methods, some statistical reference from previous studies with similar methodology should be added. For example PMID: 27005802

Answer: Thank you for your suggestion. We have edited the ‘Methods’ section in the revised manuscript to add a reference similar to this study’s method (Page 2 Line 62-63).

Reviewer 2 Report

Considering the period of 2020, I think it would be useful to have some discussions
related to the Sars-Cov2 infection in the generation of symptoms such as fever and
possibly the exclusion of those cases.

Other comorbidities that could influence the symptoms should be taken into account. The authors should specify more precisely the area of ​​origin of the patients
(hospitalized patients per ward).
The authors could add more references related to the subject. Other laboratory parameters should be discussed, such as creatinine. (cite: Serban D,
Balasescu SA, Alius C, et al. Clinical and therapeutic features of acute cholecystitis
in diabetic patients. Exp Ther Med. 2021 Jul;22(1):758. doi: 10.3892/etm.2021).
The conclusions must be more specific regarding the results obtained in the study.

Author Response

Response to Reviewer 2

We want to thank the editor and reviewers for their helpful suggestions. We have revised the manuscript and hope that this revised version has met the publication criteria. Changes were marked with a track change function in MS Word.

Considering the period of 2020, I think it would be useful to have some discussions related to the Sars-Cov2 infection in the generation of symptoms such as fever and possibly the exclusion of those cases.

Answer: Thank you for your suggestion. We agreed with the reviewer that SARS-CoV-2 infection might influence the clinical presentation of patients with fever and other nonspecific symptoms (such as abdominal pain). We have previously indicated in the limitation paragraph that the data collection duration overlapped the onset of the COVID-19 pandemic and may cause some potential bias to the clinical features and abnormal laboratory values. As a result, we reviewed patients in our cohort who presented during 2020 and had not found any positive COVID-19 cases in our cohort. We have revised this issue in the limitation paragraph in the ‘Discussion’ section (Page 11 Line 318-327).

Other comorbidities that could influence the symptoms should be taken into account.

Answer: We agreed with the reviewer on this point. Since this study was conducted retrospectively, some possible data that might have impacted the diagnosis, such as an underlying disease, was not collected. For example, patients with diabetes tended to have more severe clinical features than non-diabetic patients. Besides, the previous article focusing on clinical features of acute cholecystitis found that serum creatinine more than 2 mg/mL were observed in diabetic patients than those with non-diabetes. We have revised this point in the limitation paragraph in the ‘Discussion’ section (Page 10 Line 295-301).

The authors should specify more precisely the area of ​​origin of the patients

(hospitalized patients per ward). The authors could add more references related to the subject. Other laboratory parameters should be discussed, such as creatinine. (cite: Serban D, Balasescu SA, Alius C, et al. Clinical and therapeutic features of acute cholecystitis in diabetic patients. Exp Ther Med. 2021 Jul;22(1):758. DOI: 10.3892/etm.2021).

Answer: Our hospital represents a tertiary hospital consisting of 1500 patient beds. Of all, the general surgical ward where acute cholecystitis patients are admitted has up to 200 beds. In addition, we have discussed the performance of other laboratory parameters, such as serum creatinine, in the revised manuscript (Page 2 Line 66-67 and Page 10 Line 295-301).

The conclusions must be more specific regarding the results obtained in the study.

Answer: Thank you for highlighting this important point. With an acceptable range of diagnostic performance, we recommend using this simple score for timely diagnosis and early management for patients suspected of acute cholecystitis, especially in a setting with limited further investigations. Patients with an ED Chole Score of 0-4 could be managed considerately. In contrast, patients with an ED Chole Score of more than eight should be further investigated for acute cholecystitis. In addition, it is advised that a prospective external validation study with a larger sample size be conducted before applying this score in clinical practice. We have revised the ‘Conclusion’ section in the revised manuscript (Page 11 Line 329-338).   

Reviewer 3 Report

I read with interest authors' work. the authors had made a hard work to summarize some of the immense literature on the Diagnosis of Acute Cholecystitis. This is indeed an intriguing paper even if i can't find a real novelty. Sample sieze such as statistical analisis appear adequate (even though you might expect more patinets considering 10 ys of observative time...). There are obvious issues with selection bias, but I believe serve to better illustrate the remarkable outcomes here reported. I suggest to emprove discussion section, maybe adding someting about complications of cholecystectomy (if you want you can add the citation doi:10.23750/abm.v92iS1.10821). English might be improved, even if it is acceptable in its current form

Author Response

Please find it in the attachment.

Round 2

Reviewer 1 Report

The manuscript is OK now, thank you!

Reviewer 2 Report

The authors improved the manuscript, addressing all the queries.

Reviewer 3 Report

corrections submitted appear adequate. in my opinion the paper could be accepted for pubblication in its present form